# The Effects of Grape Polysaccharides Extracted from Grape By-Products on the Chemical Composition and Sensory Characteristics of White Wines

**DOI:** 10.3390/molecules27154815

**Published:** 2022-07-27

**Authors:** Silvia Pérez-Magariño, Estela Cano-Mozo, Marta Bueno-Herrera, Diego Canalejo, Thierry Doco, Belén Ayestarán, Zenaida Guadalupe

**Affiliations:** 1Laboratorio de Enología, Instituto Tecnológico Agrario de Castilla y León, Ctra. Burgos Km 119, 47071 Valladolid, Spain; ita-canmozes@itacyl.es (E.C.-M.); bueherma@itacyl.es (M.B.-H.); 2Departamento de Agricultura y Alimentación, Instituto de Ciencias de la Vid y el Vino (Universidad de La Rioja, Gobierno de La Rioja, CSIC), Finca de La Grajera, Ctra. Burgos 6, 26007 Logroño, Spain; diego.canalejo@unirioja.es (D.C.); belen.ayestaran@unirioja.es (B.A.); zenaida.guadalupe@unirioja.es (Z.G.); 3UMR 1083 Sciences pour l’Oenologie, INRA, SupAgro, 2 Place Viala, 34060 Montpellier, France; thierry.doco@inrae.fr

**Keywords:** wine waste valorization, grape by-products, white wines, polysaccharides, volatiles, phenols, sensory attributes

## Abstract

There is an increasing interest in the valorization of wine waste by-products. Grape pomace/marc can be an important source of polyphenols but also of polysaccharides (PSs). Therefore, the aim of this work was to extract PSs from grape pomace and musts and incorporate them into wines to improve their quality and valorize these residues. Two white wines were elaborated and treated with four different PS extracts obtained from white grape pomace, white must, a wine purified extract rich in RG-II, and commercial inactivated yeasts. In general, the use of grape PSs extracted from grape pomace or must improve some characteristics of wine, increasing the polysaccharide and volatile concentrations. These PS extracts can be useful to modulate some taste attributes such as an excess of acidity and bitterness and can also prevent the loss of volatile compounds associated with fruity and floral notes over time. This is the first study that shows the effects of grape polysaccharides on the chemical composition and sensory characteristics of white wines. Considering the obtained results, the grape pomace and surplus of musts can be considered valuable sources to obtain polysaccharide-rich products, opening a new opportunity to take advantage of by-products from the wine industry.

## 1. Introduction

The wine industry produces a large number of by-products, the skins and pomace being one of the main residues that are generated [1,2], and represents an important environmental issue. There is an increasing interest in the valorization of these by-products because different bioactive compounds can be obtained from them. The most studied compounds have been polyphenols because of their antioxidant capacity, health effects, and other technological uses in the food and winery industry [3,4], as well as grape seed oils. However, grape pomace/marc can also be an important source of polysaccharides, opening the way to a new type of exploitation of these by-products since they are currently little studied [5,6].

Polysaccharides are one of the main compounds found in wines, and they are mainly derived from the cell walls of the grape and from yeasts. The main polysaccharides in wines are polysaccharides rich in arabinose and galactose (PRAG), rhamnogalacturonans I and II (RG-I and RG-II), and homogalacturonans (HGs), which are derived from grapes, as well as mannoproteins (MPs) and glucans (GPs) that can be released by yeasts during the alcoholic fermentation process and during aging on lees [7].

Wine polysaccharides play important roles in the technological and sensory characteristics of wines. However, not all have the same behavior, and it will depend on the size and the origin of these compounds. Some studies found that PRAG had a great influence on filtration processes [8], while RG-II accelerated tartrate crystallization processes at low concentrations and inhibited them at high concentrations [9]. Other authors observed that RG-II, MPs, and PRAG could influence tannin aggregation [10,11,12,13], and therefore they would influence the body, structure, and mouthfeel of wines [14,15].

Yeast polysaccharides have been the most studied polysaccharides, especially MPs. Different authors found that MPs were more efficient in improving the tartaric stability [16,17,18,19] and protein stability of white wines [17,19,20,21,22,23,24].

Different positive effects of MPs were described in white and red wines related to sensory characteristics such as an increase in wine body and roundness in the mouth, a reduction in the astringency and bitterness sensation of the tannins [13,14,25,26,27], an improvement in wine aroma intensity, complexity and persistence [14,28,29,30,31,32], and color stabilization [25,27]. However, the desired positive effects were not always found [33,34,35] since it may depend on the commercial product associated with the mannoprotein content and purity [24,36,37].

As discussed earlier, some studies have suggested that oligosaccharides and polysaccharides from grapes (PRAG and RG-II) can significantly influence the sensory, technological, and physical–chemical characteristics of wines. The structure and characteristics of these wine oligosaccharides and polysaccharides suggested that these compounds might play important roles in the quality of wines, especially in the aggregation with tannins, color, and reducing acidity and astringency [38]. In addition, they might also indirectly influence other characteristics of wines such as aroma, protein stability, etc.

Currently, on the market, there are only commercial products rich in polysaccharides derived from yeasts, mainly mannoproteins, which have not always shown a clear effect in wines. Therefore, considering that much of the waste obtained in the wine sector is not used and that some recent studies have shown the great potential of other types of polysaccharides and oligosaccharides from grapes, the aim of this work was to extract polysaccharides (PSs) from grape pomace and musts and incorporate them into wines to improve their quality and valorize these residues. The effects of the addition of different polysaccharides extracted for grape pomace by-products and musts on the sensory and chemical composition of white wines were studied and compared with the use of commercial inactivated yeasts.

## 2. Results and Discussion

### 2.1. Characterization of Polysaccharide Extracts

Initially, the different polysaccharide extracts used were characterized, and their polysaccharide composition and molecular weight distribution are shown in Table 1. The highest total polysaccharide content was found in the wine-purified polysaccharide extract (WPP) probably due to the purification steps. The WPP had a high concentration of RG-II, which represented the 77% of total polysaccharides. The PS extract obtained from white must (WM) had a total polysaccharide content of 681 mg/g, mainly mannans (40%), GPs (27.9%), and PRAG (19%). The presence of mannose in WM was associated with the mannans of the grape pericarp, and GPs were attributed to celluloses and hemicelluloses of the cell walls [39,40].

The CIY wines presented a similar polysaccharide concentration to WM but with a different proportion of polysaccharide families, namely MPs with 69% as the most important, followed by GPs with 31%. PRAG, RG-II, and HGs were not detected in the CIY variety, as was expected because this product is a specific, inactivated yeast obtained from oenological yeasts of *Saccharomyces cerevisiae*.

The PS extract obtained from white grape pomace (WGP) showed the lowest polysaccharide concentration, even lower than those obtained in other studies [39], which could be due to the effect of the grape maturity degree and variety. The extractability of polysaccharides increased with grape maturity [41] and, with the same degree of ripening, the extractability of polysaccharides depended on the grape variety used [42].

The WGP extract was mainly formed by GPs (66.1%) and PRAG (12.6%). As in the WM extract, GPs were attributed to celluloses and hemicelluloses of the cell walls of the grape pomace [39,40]. In the literature, few studies found extracted pectin or cell-wall material from grapes, and they mainly focused on the characterization of grape cell-wall composition and structure, and, in most cases, they used toxic and stronger extractants such as ammonium oxalate, nitric acid, or sulfuric acid [43,44,45]. However, our research aimed to obtain polysaccharide extracts that could be used in the food industry. Therefore, food-grade reagents had to be used, although PSs with different characteristics and yields were obtained.

Considering the molecular weight distribution, the WGP and WM presented the highest content of HMWP, while the WPP and CIY varieties had the highest content of MMWP and LMWP, respectively.

### 2.2. Effects of the Addition of Different Polysaccharide Extracts on Chemical Composition of the Wines

Table 2 shows the oenological parameters of the initial white wines. The values of these parameters were within the values usual found in wines, with slightly higher values of total acidity.

Considering the high number of analyzed compounds (a total of 78 compounds), the individual phenolic and volatile compounds belonging to the same chemical family were grouped as follows: the phenolic compounds in hydroxybenzoic acids (HBAs), hydroxycinnamic acids (HCAs), tartaric esters of hydroxycinnamic acids (TE-HCAs), flavanols (FLAVs), tyrosol (TYR) and tryptophol (TRYP), and the volatile compounds in higher alcohols (HAs), ethyl esters of straight-chain fatty acids (EE-SCFAs), ethyl esters of branched-chain fatty acids (EE-BCFAs), alcohol acetates (AAs), terpenes (TERPs), fatty acids (FAs), C6 alcohols (C6As), vanillin derivates (VANs), and volatile phenols (VFs). These compound groups were used in subsequent statistical analyses.

Table 3 shows the concentrations of the different compounds evaluated in terms of the different experiences of the two varietal white wines and the results of ANOVA and Fisher’s least significant difference test; as it can be seen, most of the compounds showed statistically significant differences relative to treatment.

In general, as can be expected, the addition of polysaccharide extracts increased the total polysaccharide content in these wines, mainly in the wines treated with polysaccharides extracted from must (WM), followed by the wines treated with polysaccharides extracted from grape pomace (WGP). These polysaccharides mainly correspond to high-molecular-weight and medium-molecular-weight polysaccharides, between 700 and 15 kDa. These results agreed with the polysaccharide concentration of each extract. An exception was CIY wines, which showed similar or lower concentrations than the WGP and WPP wines, although the CIY extract had a higher polysaccharide concentration than WGP and WPP extracts, considering that the dose of WPP used was half of that (0.1 g/L). These results indicated that the total polysaccharides of CIY product were not solubilized in the wine, and probably, they were eliminated in the precipitate, or they could have reacted with other wine compounds without contributing to the total polysaccharide content [33]. Therefore, the studied CIY product did not provide the expected polysaccharide content.

No statistically significant differences were found in total polyphenols (TPs) and TTs between the wines treated with WM and WGP and their respective control wines (Table 3). Considering the individual phenolic compounds, the differences between WM and WGP wines and control wines were lower than 5%, in both varietal wines, with the exception of the *Albillo* wines, which showed a reduction of 12% in HBA concentrations. Therefore, the polysaccharide extracts obtained from grape and must (WGP and WM) did not affect the phenolic contents, or they were very small.

On the other hand, some differences were found in the effect of WPP and CIY extracts in the total and individual phenolic compounds (Table 3), and they depended on the varietal wine (i.e., the initial characteristics of wine). In general, these extracts reduced the content of phenolic compounds, with the exception of TPs in the *Albillo* wines. Significant differences were found in individual phenolic compounds and TTs in both varietal wines. The addition of WPP and CIY reduced the TT content in both wines, with a higher percentage of reduction in the *Verdejo* wines (33% and 46%, respectively) than in the *Albillo* wines (<14%). The same tendency was also observed in the individual phenolic compounds, mainly in HBAs and HCAs. The V-WPP wines had a loss of 68% in HBAs and 36% in HCAs, and the V-CIY wines yielded a loss of 81% in HBAs and 43% in HCAs, while the respective *Albillo* wines showed mean reductions of 19% in HBAs and 3% in HCAs contents. The losses found in TE-HCAs and FLAVs were lower (< 11% in the V-WPP wines, < 24% in the V-CIY wines, and < 4.8% in the A-WPP and A-CIY wines). The *Verdejo* wines presented higher initial contents of TPs, TTs, HBAs, and HCAs than the *Albillo* ones, which can be related to the greater effect of some polysaccharide extracts. The effects of the CIY extract can be due to the adsorption of some phenolic compounds on the yeast cell walls, results that were also found by other authors [24,37,46,47], although they were dependent on the type of commercial yeast preparation. The WPP extract was mainly composed of oligosaccharides rich in rhamnose and arabinose (RG-II) and MPs, which seemed to adsorb phenolic compounds, mainly in those wines with higher phenolic content. Both products, CIY and WPP, presented the highest percentage of LMWP and MMWP, respectively (Table 1, molecular weight < 30 kDa), which seemed to be correlated to the capacity to adsorb phenols.

No research studies have been found that use pure extracts of RG-II or grape polysaccharide extracts, so it would be interesting to carry out more studies on the interaction of this type of polysaccharides with different phenolic compounds since they could be different from those found with mannoproteins due to their different composition and structural conformation [7].

The differences observed in the volatile composition depended on the polysaccharide extract, the varietal wine, and the volatile family group. As in the phenolic composition, the *Verdejo* wines showed the highest differences. The WPP and CIY products mainly reduced the content of EE-SCFAs, AAs, TERPs, and VFs of the *Verdejo* wines and increased the content of EE-BCFAs. The *Verdejo* wines treated with these products had a mean reduction of 24% in EE-SCFAs, 46% in AAs, 69% in TERPs, and 42% in VFs compared with the control wines. As in the phenol contents, the WPP and CIY samples interacted with some volatile compounds, reducing their concentration probably due to their adsorption on these polysaccharide extracts over the contact time. These interactions were also observed by Comuzzo et al. [48] and Del Barrio-Galán et al. [31], who used different yeast derivative products in wines. However, these results were not found in all the treated wines, and it seems that the retention capacity of each yeast product depended on its composition, the macromolecule conformation, and its ability to have hydrophobic interactions with each volatile compound [32]. On the other hand, in general, the WM and WGP extracts did not produce significant changes in the volatile composition of the *Verdejo* wines.

In the case of the *Albillo* wines, the WM, WGP, and WPP wines showed higher concentrations of some volatile compounds than the control wines, such as in EE-SCFAs, AAs, TERPs, and C6As. These results indicated that these PS extracts could reduce the hydrolysis processes of EE-SCFAs and AAs and/or modify the hydrolysis/esterification balance between ethyl esters and alcohols that naturally occur during the aging or storage period [49,50]. Therefore, these products can favor the maintenance of volatile compounds associated with fruity and floral notes over time. On the other hand, the CIY variety reduced the content of the EE-SCFAs, TERPs, and VFs, as was also shown in the *Verdejo* wines.

### 2.3. Multivariate Statistical Analyses

PCA was performed to investigate whether the information given by the significant compounds would allow differentiating the white wines by the addition of polysaccharide extracts.

The PCA of the *Albillo* wines selected four components with an eigenvalue greater than 1, which explained 95.3% of the total variance. Figure 1a shows the distribution of the different treated *Albillo* wines in the plane defined by the first two principal components (PCs), which explained 67.1% of the total variance. The variables associated with the two PCs allow differentiating the wines by treatment. The A-C wines are located in the upper left part of the plane, the A-CIY wines in the lower left part, and the others in the right part of the plane. The A-WPP and A-CIY wines presented the greatest distances from the A-C wines, followed by the A-WM wines. Considering the variables associated with these PCs (Figure 1b), the A-CIY wines presented higher color intensity and TE-HCAs, and lower total tannins (TTs), TRYP, and FLAVs than the control wines. On the other hand, A-WGP and A-WM were characterized by high TTs, TRYP, HCAs, EE-SCFAs, TERP, AAs, and polysaccharide contents, while the A-WPP variety, by high AAs and EE-SCFAs and low HBAs, HCAs, and VANs.

In the case of the *Verdejo* wines, the PCA selected only two components with an eigenvalue greater than 1, which explained 89.8% of the total variance. These wines are represented in Figure 2a and are clearly differentiated between them. The V-C wines are in the lower part, while the V-WM and V-WGP varieties are sited in the upper right part and the V-CIY and V-WPP wines in the upper left part.

As in the *Albillo* wines, the V-WPP, V-CIY, and V-WM wines were the ones that showed more differences with respect to the V-C wines. All the phenolic compounds, color, and most of the volatile compounds (EE-SCFAs, AAs, VFs, TERP, and C6As) were strongly and positively correlated with PC1 (Figure 2b), and the concentrations of these compounds were high in the V-WM and V-WGP wines. The V-WM wines also presented the highest polysaccharide concentration, followed by the V-WGP wines. On the other hand, the V-CIY and V-WPP varieties presented a lower content of phenolic and volatile compounds, with the exception of TYR and EE-BCFAs (Table 3).

### 2.4. Effects of the Addition of Different Polysaccharide Extracts on Sensory Characteristics of the Wines

Figure 3 provides a GPA consensus configuration for the *Albillo* wines as determined in terms of their olfactory and gustative attributes. The olfactory GPA space defined by the two first factors accounted for 70.6% of the total variance (Figure 3a), and the treated wines were clearly different from the control wines. The A-C wines were characterized by yeasty, floral, and herbaceous notes, while the A-WM and A-WGP wines presented high tropical and stone fruity notes, and the A-CIY and A-WPP wines stood out for their great notes of white fruit and lees. Figure 3b shows the GPA average space obtained from the gustative attributes, which explained 64.8% of the total variance. The A-C wines were characterized by higher acidity and lower persistence and balance than the treated wines. The A-WGP and A-CIY wines showed higher correlations with persistence than the other wines, while the A-WM wines were more correlated with body and the A-WPP samples with high sweetness and low bitterness.

The GPA consensus plot for the *Verdejo* wines in terms of the olfactory and gustative attributes also differentiated the control wines from the treated ones (Figure 4). The olfactory GPA space defined by the two first factors accounted for 86.6% of the total variance (Figure 4a). The V-C wines were correlated with white fruity, stone fruity, and herbaceous notes. The V-WGP and V-WM wines presented similar olfactory characteristics that were correlated with tropical fruity notes, while V-WPP and V-CIY were highlighted by yeasty and lees notes. Figure 4b represents the gustative GPA space of the *Verdejo* wines, which explained 72.9% of the total variance. The V-C wines were characterized by acidity, bitterness, and body, while the V-WPP and V-CIY varieties, by sweetness. However, the V-WGP and V-WM wines did not emphasize any particular gustative attribute.

Some of the observed chemical changes were correlated with the sensory characteristics of the wines. In the olfactory phase, the *Albillo* wines treated with WM and WGP extracts were characterized by tropical and stone fruity notes, associated with the highest concentrations of EE-SCFAs, AAs, and TERP, while the CIY and WPP extracts provided high lees notes. In the *Verdejo* wines, the control wines were more associated with all fruity notes, and the wines treated with CIY and WPP were associated with high lees and yeasty notes and low fruity aromas. Del Barrio-Galán et al. [51] showed similar results in *Verdejo* wines treated with different commercial yeast products after six months of treatment, while Bautista et al. [52] showed a decrease in floral and fruity notes after seven months of aging in a white wine made from a mixture of Galician grape varieties. However, all the treated wines showed lower olfactory intensity than the control wines, which might be related to the lower volatility of aroma compounds. Some studies have reported that polysaccharides can reduce the volatility of wine flavor compounds, which has been associated with an increase in hydrophobic interactions between the aroma compounds and proteoglycans [53].

Regarding taste attributes, in general, all the products reduced the sensation of acidity and bitterness, and the WPP and CIY products gave rise to sweeter wines, both in the *Albillo* and *Verdejo* wines. No clear effect of the PS extracts on body and persistence was found. The reduction in bitterness could be due to all the main polyphenols found in white wine, which can form hydrogen-bonded complexes with polysaccharides [54], and these complexes may hinder the interaction between bitter phenols with taste receptors through steric effects. Different results have been found in the bibliography. Gawel et al. [38] did not find a reduction in bitterness by adding polysaccharides in white wines with high phenol content. However, Vidal et al. [26] and Del Barrio-Galán et al. [51] found a decrease in bitterness due to the addition of a mixture of MPs and PRAG (neutral polysaccharide fraction) and commercial yeast derivatives mainly composed of MPs, respectively.

On the other hand, contradictory results have been found in the bibliography related to perceived acidity and sweetness. The increase in the sweetness of wines treated with WPP and CIY could be due to the higher content of LMWP and MMWP in these polysaccharide extracts (Table 1), which can provide a high-sugar characteristic. In addition, some authors found that RG-II (MMWP) was more effective to reduce protein–tannin interactions between wine polyphenols and saliva protein than PRAG [55], and therefore, it could be used to modulate astringency and indirectly influence sweetness. Similar results were found by Del Barrio-Galán et al. [51], who used commercial yeast derivatives rich in MPs, while Gawel et al. [38] showed that the addition of polysaccharides did not affect acidity and sweetness in white wines.

The mouthfeel, viscosity, and body are sensory attributes difficult to understand because there are many compounds such as ethanol, glycerol, polyphenols, and polysaccharides that can play different roles in contact with the mouth surfaces [56]. Although polysaccharides seem to play important roles in the sensory body attribute, the addition of polysaccharides did not always improve this attribute, which could be due to the characteristics and composition of the initial wine also affecting the perception of the body. Gawel et al. [38] indicated that wine pH and ethanol had more effects on mouthfeel and taste than polysaccharides. However, they showed that, at high pH (3.6), the MMWP fraction (13–93 kDa) caused a significant increase in the perceived body, while the LMWP fraction (5–13 kDa) did not affect this attribute in white wines. These authors hypothesized that larger PSs with a high charge density favored hydration with unbound free water, reducing electrostatic effects and increasing the body perception. Vidal et al. [26] and Del Barrio-Galán et al. [51] also found an increase in body perception by the addition of polysaccharides rich in mannoproteins in model wine solutions and white wines, respectively.

## 3. Materials and Methods

### 3.1. Polysaccharide Extractions

White pomace was obtained from the *Verdejo* grape variety after pressing. Polysaccharides from white grape pomace were extracted following the methodology previously developed by our group [39]. Briefly, defrosted pomace was homogenized using an Ultra-Turrax, and the extraction with tartaric acid was carried out for 18 h in an orbital shaker. Then, the samples were centrifuged, and the supernatants were concentrated five times in a rotavapor. Polysaccharides were precipitated with four volumes of cold acidified ethanol for 20 h at 4 ºC. These precipitates were dissolved in Milli-Q water and freeze-dried to obtain the white grape pomace polysaccharides (WGPs).

Commercial white concentrated must (65 ºBrix) was used and diluted 1:1 with Milli-Q water. Polysaccharides from must were recovered via precipitation with cold acidified ethanol for 20 h at 4 ºC [57]. These precipitates were dissolved in Milli-Q water and freeze-dried to obtain the wine must polysaccharides (WMs).

The wine-purified polysaccharides (WPPs) were obtained from the freeze-dried polysaccharides extracted from *Carignan noir* wine following the method described in Canalejo et al. [40]. This extract was rich in rhamnogalacturonans type II.

### 3.2. Winemaking and Treatments

White wine elaborations were carried out in the experimental winery of the Oenological Station (ITACyL) sited in Rueda (Valladolid, Spain). Two white wines from *Verdejo* and *Albillo* grape varieties were elaborated in stainless steel tanks of 150 L in duplicate following the traditional white winemaking process. The grapes were manually harvested, and transported to the Oenological Station in 15 kg plastic boxes, and they were destemmed, crushed, slightly sulfited (0.05 g/L), and pressed. The obtained musts were transferred to stainless steel tanks, and a pectinolytic enzyme preparation was added (0.01 g/L of EnartisZym RS, Enartis, Italy) to favor the precipitation of colloidal substances over 24–48 h at 7 ºC. After that, the musts were racked off into different stainless-steel tanks and inoculated with commercial *Saccharomyces cerevisiae* yeasts (0.20 g/L of EnartisFermPerlage, Enartis, Italy) to undergo alcoholic fermentation at a controlled temperature (16-18 ºC). Once alcoholic fermentation was completed, the wines were kept in the tanks for 4 days, and after this time, they were racked off.

Five experiments were carried out with each wine in duplicate: control wines (without the addition of any product, C); wines with the addition of PSs extracted from WGP; wines with the addition of PSs extracted from WM; wines with the addition of WPP; and wines with the addition of commercial inactivated yeasts (CIY, Lallemand, Logroño, Spain). The doses used for the different polysaccharide extracts were 0.20 g/L, with the exception of the WPP variety, which was 0.10 g/L. These products were maintained in contact with the wines for two months at 15 ± 1 ºC, and two batonnage procedures were performed per week on each wine. After this time, the wines were filtered and bottled and kept at a controlled temperature (14 ºC ± 1 ºC) for six months until analyses.

### 3.3. Standards, Gases, and Chemical Reagents

The volatile and phenolic compound standards were purchased from Fluka (Buchs, Switzerland), Sigma-Aldrich (Steinheim, Germany), Alfa Aesar (Lancashire, United Kingdom), and Extrasynthese (Genay, France).

Helium BIP (99.9997%), air zero (99.998%), and Premier plus hydrogen (99.9992%) were provided by Carburos Metálicos S.A. (Valladolid, Spain).

Food-grade reagents were used for polysaccharide extractions: hydrochloric acid 37% (E-507, Panreac, Madrid, Spain), tartaric acid (E-334, Agrovin, Ciudad Real, Spain), and rectified alcohol from molasses 96º (0110F, Alcoholes Montplet, Barcelona, Spain).

Chromatographic-grade reagents were provided by Riedel-de-Haën (Honeywell, Germany), and the remaining reagents were supplied by Panreac (Madrid, Spain). Milli-Q-grade water was obtained using a Millipore system (Bedford, MA).

### 3.4. Analytical Methods

The monosaccharide composition of the extracts was determined via gas chromatography with a mass detector (GC-MS) of their trimethylsilyl-ester-O-methyl glycosyl derivates, following the methodology described by Guadalupe et al. [57], determining the polysaccharides rich in arabinose and galactose (PRAG), rhamnogalacturonans type II (RG-II), homogalacturonans (HGs), mannans and mannoproteins (MPs), and glucosyl polysaccharides (GPs). Total polysaccharides (TPSs) were calculated as the sum of all of them. In addition, the molecular weight distributions of the extracts were estimated via high-performance size exclusion chromatography coupled to a refractive index detector (HPSE-RID) following the methodology described in Guadalupe et al. [57].

Standard oenological parameters in wines were determined according to the official analysis methods of the OIV: pH, titratable acidity (as g/L tartaric acid), SO_2_, and alcohol content (%vol: mL ethanol/100 mL wine) [58]. Malic and acetic acids were analyzed using enzymatic kits.

Color intensity was determined by absorbance measurement at 420 nm, and the total content of phenolic compounds was evaluated by the quantification of total polyphenols (TP, mg/L of gallic acid) and total tannins (TT, mg/L of cyanidin chloride). All of these parameters were measured using a UV/Vis Agilent Cary 60 spectrophotometer (Santa Clara, CA, USA) and 1 cm quartz cuvettes.

Low-molecular-weight phenolic compounds were analyzed via direct injection in an Agilent Technologies LC-DAD series 1200 chromatograph (Waldbronn, Germany) coupled to a diode array detector (DAD), following the chromatographic conditions described by Pérez-Magariño et al. [59]. Wine samples were previously diluted in water (1:1) and filtered through PVDF filters of 0.45 μm. Seventeen phenolic compounds were quantified and expressed in milligrams per liter of the corresponding phenolic standard.

Higher alcohols were analyzed via direct injections of the wines, following the method described in Pérez-Magariño et al. [60], using a gas chromatograph with a flame ionization detector (GC-FID). Minor volatile compounds were extracted via headspace solid-phase microextraction (HS-SPME), following the methodology described in Del Barrio-Galán et al. [61], and were quantified in a GC-MS, following the chromatographic conditions established by Pérez-Magariño et al. [60]. Fifty-four volatile compounds were quantified and expressed in milligrams per liter of the corresponding volatile standard.

Total wine polysaccharides (T-PS) were estimated by using HPSEC–RID following the methodology described in Guadalupe et al. [57]. Polysaccharide contents were estimated using calibration curves constructed from dextran standards from 410 to 5 kDa and were expressed in milligrams per liter of dextrans. Three different polysaccharide fractions were estimated according to their molecular weight: high-molecular-weight polysaccharides (HMWPs: 700–100 kDa), medium-molecular-weight polysaccharides (MMWPs: 100–15 kDa), and low-molecular-weight polysaccharides (LMWPs: 3–15 kDa). The polysaccharides evaluated by using HPSEC-RID have a good correlation with the polysaccharides evaluated by using GC-MS, and the process is more rapid and simple and, therefore, can be useful for estimating the content of T-PS [57].

### 3.5. Sensory Analysis

The sensory analysis was performed in a designed test room in accordance with ISO 8589 Standard (2010) and was carried out by 7 expert tasters who were trained to quantify the descriptors previously defined using a structured numerical scale of five points. Samples were presented in standard glasses in random order and codified with a three-digit number.

### 3.6. Statistical Analyses

A one-way analysis of variance (ANOVA) test and Fisher’s least significant difference (LSD) test were performed using the Statgraphics Centurion XVIII statistical package (Statgraphics Technologies, Inc., Virginia, USA) to determine the effects of treatments. Principal component analysis (PCA) was carried out with the significant variables to study their association and to determine similarities or differences between wines relative to treatment, using the RStudio program. Generalized Procrustes analysis (GPA) was applied to the values of the olfactory and gustative attributes using the XLSTAT 2022.1 Software (Addinsoft Inc, New York, USA).

## 4. Conclusions

No studies using polysaccharides extracted from grape (pomace and/or musts) or purified extracts rich in RG-II have been found, and only some studies used purified grape pomace as a fining agent in red wines [62] and, in general, in the food industry with different technological functions [6]. Therefore, this is the first study showing the effects of grape polysaccharides on the chemical composition and sensory characteristics of white wines. In general, the use of grape polysaccharides extracted from grape pomace or must improved some wine characteristics, such as polysaccharide and volatile composition, increasing their concentrations. Grape polysaccharide extracts from grape pomace and must can also be useful to modulate some taste attributes such as an excess of acidity and bitterness and increase the body and persistence of wine products to a similar or greater degree than commercial yeast-derived products (rich in mannoproteins). In addition, they can prevent the loss of volatile compounds associated with fruity and floral notes over time, which will have a significant sensory impact on olfactive attributes. These results can be associated with the PS type since the WGP and WM samples were richest in high-molecular-weight PSs (glucosyl PSs from celluloses and hemicelluloses and PRAG).

Therefore, the grape pomace and surplus of musts can be considered valuable sources to obtain polysaccharide-rich products to be used as winemaking additives, opening a new opportunity to take advantage of by-products from the wine industry. Furthermore, it would be interesting to improve the extractability and purity of the polysaccharide extracts obtained from these by-products.

As the addition of polysaccharides extracted from grape by-products to white wines modified their chemical composition and sensory characteristics, but not always in the same way, it would also be interesting to study the effect on different wines in order to determine what causes the greater or lesser interaction of the polysaccharide extracts with different chemical compounds, mainly phenols and volatiles.

## Figures and Tables

**Figure 1 molecules-27-04815-f001:**
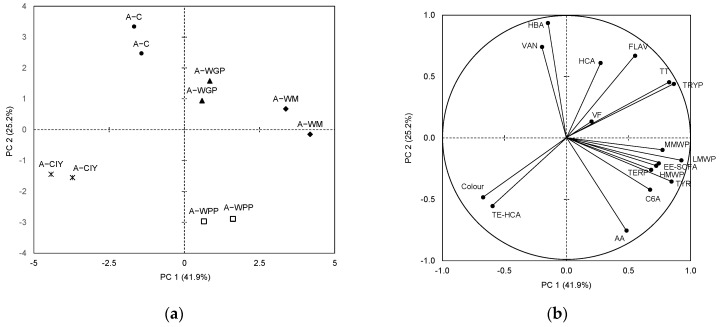
Principal component analysis: (**a**) distribution of the *Albillo* wines; (**b**) loadings of the variables. C: control wines, WM: wines with the addition of PSs extracted from white must, WGP: wines with the addition of PSs extracted from white grape pomace, WPP: wines with the addition of wine purified polysaccharide rich in RG-II, CIY: wines with the addition of commercial inactivated yeasts. Abbreviations of compounds in Materials and Methods section.

**Figure 2 molecules-27-04815-f002:**
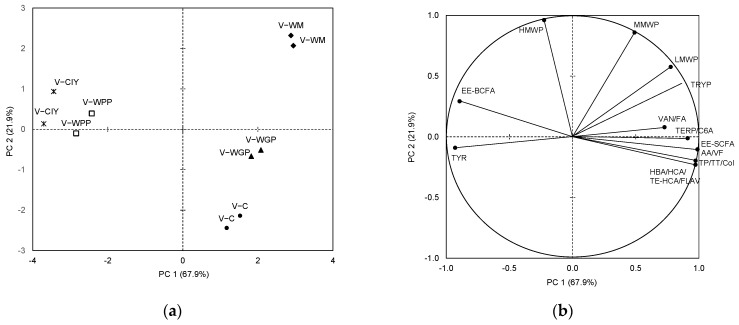
Principal component analysis: (**a**) distribution of the *Verdejo* wines; (**b**) loadings of the variables. C: control wines, WM: wines with the addition of PSs extracted from white must, WGP: wines with the addition of PSs extracted from white grape pomace, WPP: wines with the addition of wine purified polysaccharide rich in RG-II, CIY: wines with the addition of commercial inactivated yeasts. Abbreviations of compounds in Materials and Methods section.

**Figure 3 molecules-27-04815-f003:**
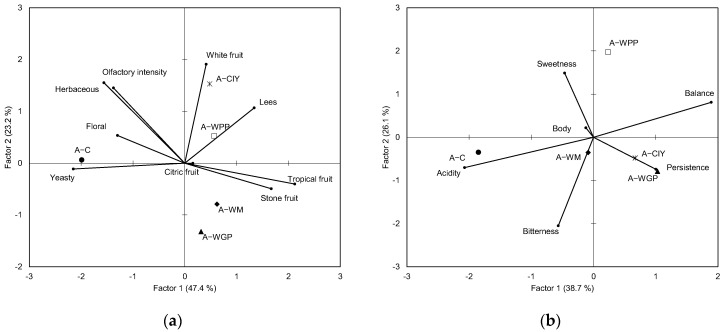
Generalized Procrustes analysis (GPA) of the mean ratings for (**a**) olfactory phase and (**b**) gustative phase in the *Albillo* wines. C: control wines, WM: wines with the addition of PSs extracted from white must, WGP: wines with the addition of PSs extracted from white grape pomace, WPP: wines with the addition of wine purified polysaccharide rich in RG-II, CIY: wines with the addition of commercial inactivated yeasts.

**Figure 4 molecules-27-04815-f004:**
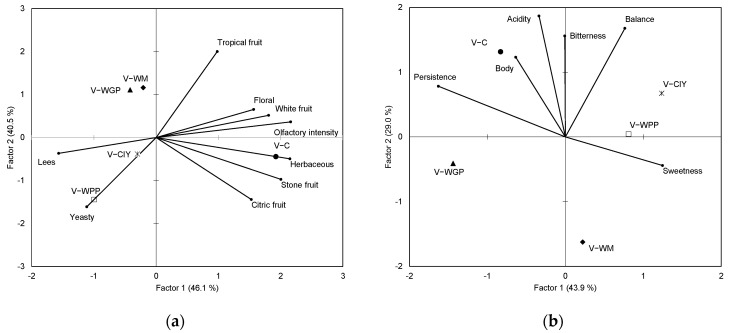
Generalized Procrustes analysis (GPA) of the mean ratings for (**a**) olfactory phase and (**b**) gustative phase in the *Verdejo* wines. C: control wines, WM: wines with the addition of PSs extracted from white must, WGP: wines with the addition of PSs extracted from white grape pomace, WPP: wines with the addition of wine purified polysaccharide rich in RG-II, CIY: wines with the addition of commercial inactivated yeasts.

**Table 1 molecules-27-04815-t001:** Polysaccharide concentration (mg polysaccharide/g of extract) of the extracts determined with GC-MS and standard deviation in parenthesis and molecular weight distribution (%) determined with HPSEC-RID.

Extracts^1^	PRAG^2^	RG-II^2^	HG^2^	MP^2^	GP^2^	TPS^2^	HMWP^3^	MMWP^3^	LMWP^3^
**WGP**	36.9 (5.5)	26.0 (2.8)	24.8 (5.8)	11.1 (0.8)	193 (17.9)	292 (19.8)	45.5	0.0	54.5
**WM**	129 (16.3)	33.0 (2.9)	55.7 (8.4)	273 (20.1)	190 (13.5)	681 (30.5)	44.7	45.3	10.0
**WPP**	132 (6.9)	670 (36.8)	61.8 (4.4)	3.7 (0.8)	28.8 (5.3)	897 (38.1)	0.0	71.2	28.8
**CIY**	nd ^4^	nd	nd	428 (17.7)	189 (22.4)	617 (28.6)	31.6	0.0	68.4

^1^ WGP: white grape pomace; WM: white must; WPP: wine-purified polysaccharide; CIY: commercial inactivated yeasts; ^2^ PRAG: polysaccharide rich in arabinose and galactose; RG-II: rhamnogalacturonans type II; HG: homogalacturonans; MP: mannans or mannoproteins; GP: glucosyl polysaccharides (celluloses and hemicelluloses); TPS: total polysaccharides as the sum of PRAG, RG-II, HGs, MPs, and GPs; ^3^ HMWP (high-molecular-weight PS, > 30 kDa); MMWP (medium-molecular-weight PS, 5-30 kDa); LMWP (low-molecular-weight PS, < 5 kDa); ^4^ nd: not detected.

**Table 2 molecules-27-04815-t002:** Oenological parameters of the white wines before the treatments.

Parameters	*Albillo* Wines	*Verdejo* Wines
Alcohol degree (% etanol *v*/*v*)	12.2	12.9
Total acidity (g/L of tartaric acid)	7.0	7.2
pH	3.10	3.05
Acetic acid (g/L)	0.13	0.37
Malic acid (g/L)	2.1	2.0
Total SO_2_ (mg/L)	40	54

**Table 3 molecules-27-04815-t003:** Parameters and compounds (in mg/L) analyzed of the *Albillo* and *Verdejo* wines^1^.

	*Albillo* Wines					*Verdejo* Wines				
	A-C	A-WM	A-WGP	A-WPP	A-CIY	V-C	V-WM	V-WGP	V-WPP	V-CIY
Colour	0.075±0.001 b	0.077±0.001 c	0.067±0.001 a	0.078±0.001 c	0.101±0.001 d	0.113±0.001 c	0.118±0.001 d	0.130±0.001 e	0.092±0.001 b	0.083±0.001 a
TP	129±1.7	131±0.5	130±2.1	127±2.4	128±1.0	172±0.7 c	171±1.7 c	168±0.4 c	125±1.1 b	122±0.8 a
TT	84±0.6 bc	87±0.5 c	87±1.1 c	82±2.5 b	72±1.6 a	161±2.7 cd	170±2.9 d	157±2.5 c	107±1.9 b	87±1.9 a
T-PS	173±2.6 a	277±9.7 d	212±3.6 c	210±9.7 bc	191±7.1 b	123±2.9 a	223±4.0 c	160±5.3 b	158±6.5 b	158±9.8 b
HMWP	92.2±0.3 a	123±0.4 c	103±1.0 b	103±2.0 b	102±1.9 b	61.3±2.2 a	98.9±1.2 c	80.0±1.5 b	89.6±4.5 bc	92.8±4.1 c
MMWP	58.9±0.3 a	100±6.2 c	70.6±0.3 b	68.4±1.0 b	65.8±1.1 b	43.8±1.1 a	84.5±3.3 c	56.0±1.3 b	49.5±0.7 ab	54.3±5.1 b
LMWP	22.0±0.6 a	54.5±5.0 c	38.6±2.3 b	38.9±6.8 b	23.6±6.2 a	18.0±0.4 b	39.5±1.9 d	23.7±2.5 c	18.7±1.3 b	11.2±0.5 a
HBA	2.24±0.01 e	1.93±0.01 c	1.99±0.01 d	1.77±0.01 a	1.86±0.03 b	10.9±0.07 d	10.5±0.06 c	10.9±0.01 d	3.48±0.08 b	2.05±0.04 a
HCA	4.23±0.01 c	4.29±0.01 d	4.53±0.01 e	4.06±0.01 a	4.15±0.01 b	11.4±0.01 d	11.2±0.01 c	11.5±0.03 e	7.21±0.01 b	6.44±0.01 a
TE-HCA	3.44±0.06 b	3.28±0.06 a	3.29±0.01 a	3.53±0.01 b	3.51±0.01 b	2.85±0.04 c	2.80±0.01 c	3.00±0.06 d	2.53±0.01 b	2.16±0.03 a
TYR	26.7±0.59 a	28.2±0.10 c	27.1±0.33 ab	27.7±0.10 bc	26.7±0.17 a	27.5±0.92 ab	26.5±0.25 a	26.7±0.10 a	28.7±0.47 bc	29.5±0.10 c
TRYP	6.37±0.04 b	6.54±0.03 c	6.35±0.04 b	6.30±0.08 b	6.02±0.03 a	1.43±0.07 c	1.39±0.07 bc	1.29±0.01 b	0.66±0.03 a	0.64±0.01 a
FLAV	2.89±0.04 c	2.86±0.01 bc	2.85±0.01 bc	2.83±0.01 b	2.75±0.01 a	1.43±0.01 d	1.40±0.01 c	1.47±0.01 e	1.34±0.01 b	1.26±0.01 a
HA	336±11	341±12	341±6	330±11	332±6	272±7	271±6	268±6	281±7	283±5
EE-SCFA	3.32±0.017 b	3.49±0.027 c	3.44±0.032 c	3.57±0.011 d	3.00±0.032 a	4.13±0.028 b	4.30±0.004 b	4.12±0.069 b	3.18±0.041 a	3.12±0.017 a
EE-BCFA	0.042±0.001	0.039±0.007	0.039±0.007	0.042±0.004	0.036±0.005	0.044±0.002 a	0.046±0.001 a	0.041±0.004 a	0.054±0.001 b	0.064±0.001 c
AA	1.204±0.004 a	1.310±0.065 b	1.313±0.006 b	1.434±0.039 c	1.269±0.047 ab	1.075±0.052 b	1.103±0.029 b	1.152±0.032 b	0.596±0.025 a	0.551±0.010 a
FA	16.6±0.159	16.5±0.079	16.4±0.055	16.7±0.031	16.4±0.065	17.4±0.081 ab	17.8±0.132 b	18.3±0.201 c	17.3±0.030 a	17.4±0.038 ab
C6A	1.09±0.018 a	1.22±0.070 bc	1.25±0.015 c	1.25±0.010 c	1.13±0.033 ab	1.36±0.026 bc	1.47±0.152 c	1.46±0.014 c	1.26±0.041 ab	1.08±0.014 a
TERP	0.110±0.014 b	0.128±0.001 c	0.122±0.006 c	0.138±0.001 c	0.079±0.001 a	0.115±0.001 d	0.131±0.002 e	0.078±0.007 c	0.051±0.002 b	0.021±0.002 a
VAN	0.178±0.003 c	0.162±0.002 bc	0.142±0.024 b	0.117±0.006 a	0.154±0.001 bc	0.205±0.008 bc	0.226±0.034 cd	0.244±0.025 d	0.162±0.010 a	0.199±0.009 b
VF	0.677±0.048 c	0.691±0.046 c	0.515±0.035 a	0.604±0.022 b	0.600±0.020 b	0.967±0.019 b	1.026±0.098 b	0.970±0.047 b	0.568±0.007 a	0.537±0.003 a

^1.^ Mean values ± standard deviation. Values with different letters in each compound indicate statistically significant differences at *p* < 0.05. Abbreviations in Section 3.

## Data Availability

Not applicable.

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
