# Peer review of "The Effects of Grape Polysaccharides Extracted from Grape By-Products on the Chemical Composition and Sensory Characteristics of White Wines"

_molecules, 2022, doi:10.3390/molecules27154815_

Round 1
Reviewer 1 Report
Present manuscript aimed to extract polysaccharides (PS) from grape pomace and musts and incorporate them into wines to im prove their quality and valorize these residues. The effect of the addition of different polysaccharides extracted for grape pomace by-products and musts on sensory and chemical composition of white wines was studied and was compared with the use of commercial inactivated yeasts. In generally, the abstract was clearly and accurately describing the content of the article; The problem was significant and concisely stated. I suggest accept it after minor revisions.
1. In text, author claimed ‘Polysaccharides from white grape pomace were extracted following the methodology previously developed by our group [39]’. I suggest the physiochemical properties of Polysaccharides should be introduced in text briefly.
2. Part 4.5. Sensory analysis: author claimed ‘The sensory analysis was performed in a designed test room in accordance with ISO 8589 Standard (2010) and was carried out by 7 expert tasters who were trained to quantify the descriptors previously defined using structured numerical scale of five points.’. They evaluated the samples acceptance by using a low number of expert subjects. This is wrong from a methodological point of view. They should use a high number of subjects, e.g., 100 or more, who are consumers (not trained subjects or experts).
3. Part Conclusions:Present version like the discussion, I suggest author rewrite this part.
4. The writing English should be improved by native English speaker. I strongly suggest to check the manuscript carefully, especially please check the grammar and the completeness of the sentences once again. And please check the tense of the sentences. There should be consistency throughout the manuscript using past tense.
Author Response
Below appear an itemized list of changes and responses made to the reviewer 1.
- “In text, author claimed ‘Polysaccharides from white grape pomace were extracted following the methodology previously developed by our group [39]’. I suggest the physiochemical properties of Polysaccharides should be introduced in text briefly.”.
Response: I'm sorry, but I really don't understand what physicochemical properties the reviewer wants us to include. Canalejo et al [39] paper explains the methodology used to the polysaccharide extraction from grape pomace. The characterization of this product (WGP extract) obtained appears in Table 1.
- “Part 4.5. Sensory analysis: author claimed ‘The sensory analysis was performed in a designed test room in accordance with ISO 8589 Standard (2010) and was carried out by 7 expert tasters who were trained to quantify the descriptors previously defined using structured numerical scale of five points.’. They evaluated the samples acceptance by using a low number of expert subjects. This is wrong from a methodological point of view. They should use a high number of subjects, e.g., 100 or more, who are consumers (not trained subjects or experts).”
Response: The reviewer is right that in order to evaluate the acceptance of a product it is necessary to involve a large number of consumers. However, in this study, we have not evaluated the acceptance of these wines, we wanted to know the effects of the different polysaccharide extracts on the sensory characteristics of the wines and so the sensory analyses were carried out by expert tasters.
- “Part Conclusions Present version like the discussion, I suggest author rewrite this part.”
Response: I have read the conclusions several times and I consider that it is not like the discussion. The positive contribution that the use of these polysaccharide extracts can make to the characteristics of wine and the possibilities for future research are described. Therefore, I would be grateful if you could indicate which part you consider inappropriate to rewrite. Thank you.
- “The writing English should be improved by native English speaker. I strongly suggest to check the manuscript carefully, especially please check the grammar and the completeness of the sentences once again. And please check the tense of the sentences. There should be consistency throughout the manuscript using past tense.”
Response: We have followed your recommendation. The English has been revised and the past tense has been used.
Reviewer 2 Report
The manuscript focuses on using by-products to enhance chemical composition and sensory characteristics in wine. The concept of using by-product to minimize waste is of interest particularly since the waste by-products are known to be packed with high phytochemical/metabolite constituents with important benefits. Please see below some of my comments and recommendations:
In the results section (2.1), the authors have described their results and supplemented it with already published work. This should be corrected, either the results and discussion are combined on distinctly separated.
Statistical analysis for Table 3 is not very clear. Which type of statistical analysis was applied? Why was the SE/SD not presented in the table? Which post hoc test was used to determine the different letters? Does the P-value column represent the overall changes of the different parameters?
Since the discussion is well-prepared and points out important findings from the study, I would suggest the authors combine the results and discussion section. If not, the results section will need to be brief and improved to better understand what the authors are intending on conveying to the reader.
Author Response
Below appear an itemized list of changes and responses made to the referee 2.
- “Statistical analysis for Table 3 is not very clear. Which type of statistical analysis was applied?
Response: In the manuscript, we say “Table 3 shows the concentrations of the different compounds evaluated of the different experiences of the two varietal white wines and the ANOVA results”. So, we have applied an ANOVA. However, we have also applied the LSD test, and this fact has been included in the revised version to better understand.
- “Why was the SE/SD not presented in the table?
Response: The SD values have not been added so as not to complicate the understanding of the table and because all the data would not fit in the same table. Statistically significant differences between treatments are calculated considering the SD values, therefore the letters are indicative of them. However, if the editor and/or the referee considers necessary to include these values, we would have no objection to add them, but table 3 should be divided into two new tables.
- “Which post hoc test was used to determine the different letters?
Response: The method used to discriminate between means is Fisher's Least Significant Difference (LSD) procedure. We have included it in the revised version of the manuscript, in section 4.6. (now section 3.6).
- “Does the P-value column represent the overall changes of the different parameters?”.
Response: The p-values are determined by the ANOVA and when the p-value is less than 0.05, there is a statistically significant difference between treatments in each compound, at the 5% level of significance. This column represents this p-values by compound and in bold it showed statistically significant differences (p<0.05). However, if it is not clear, this column could be eliminated, since the letters of Fisher's test already indicate the existence of significant differences.
- “Since the discussion is well-prepared and points out important findings from the study, I would suggest the authors combine the results and discussion section. If not, the results section will need to be brief and improved to better understand what the authors are intending on conveying to the reader.”
Response: As suggested by reviewer 2, the results and discussion sections have been combined in the same section.